# Unveiling the Secrets of Escher’s Lithographs

**DOI:** 10.3390/jimaging6020005

**Published:** 2020-02-21

**Authors:** Primo Coltelli, Laura Barsanti, Paolo Gualtieri

**Affiliations:** 1Istituto Scienza e Tecnologie dell’Informazione, CNR, Via Moruzzi 1, 56124 Pisa, Italy; primo.coltelli@isti.cnr.it; 2Istituto di Biofisica, CNR, Via Moruzzi 1, 56124 Pisa, Italy; laura.barsanti@pi.ibf.cnr.it

**Keywords:** retina model, Escher, receptive fields, impossible structures, pre-attentive perception

## Abstract

An impossible structure gives us the impression of looking at a three-dimensional object, even though this object cannot exist, since it possesses parts that are spatially non-connectable, and are characterized by misleading geometrical properties not instantly evident. Therefore, impossible artworks appeal to our intellect and challenge our perceptive capacities. We analyzed lithographs containing impossible structures (e.g., the Necker cube), created by the famous Dutch painter Maurits Cornelis Escher (1898–1972), and used one of them (The Belvedere, 1958) to unveil the artist’s hidden secrets by means of a discrete model of the human retina based on a non-uniform distribution of receptive fields. We demonstrated that the ability of Escher in composing his lithographs by connecting spatial coherent details into an impossible whole lies in drawing these incoherent fragments just outside the zone in which 3D coherence can be perceived during a single fixation pause. The main aspects of our paper from the point of view of image processing and image understanding are the following: (1) the peculiar and original digital filter to process the image, which simulates the human vision process, by producing a space-variant sampling of the image; (2) the software for the filter, which is homemade and created for our purposes. The filtered images resulting from the processing are used to understand impossible figures. As an example, we demonstrate how the impossible figures hidden in Escher’s paintings can be understood.

## 1. Introduction

According to the dictionary definition, illusions are perceptions that represent what is perceived in a way different from the way it is in reality; i.e., they are instances of a wrong or misinterpreted perception of a sensory experience. Illusions can be considered powerful tools for the systematic investigation of human perception [1]. Impossible structures are illusions [2]. An impossible structure gives the onlooker the impression of looking at three-dimensional objects that cannot exist, since they possess parts that are spatially non-connectable, and are characterized by misleading geometrical properties that are not pre-attentively evident [3]. Pre-attentively means within the time pause of fixation (120 ms); any perception possible within this time frame involves only the information available in a single glimpse [4]. Examples of impossible structures are the Penrose triangle, the Penrose infinite staircase, and the Necker cube [2].

Impossible structures have drawn attention because they cause a mental conflict, thus creating emotions. Artists know this very well and utilize impossible figures to challenge the onlooker. The famous Dutch graphic artist Maurits Cornelis Escher (1898–1972) is especially well-known for the impossible structures he inserted in his lithographs, such as the Penrose triangle present in Waterfall (1961), the Penrose infinite staircase present in Ascending and Descending (1960), and the Necker cube present in Belvedere (1958) (Figure 1). Through these impossible structures, Escher challenged the concept of the real world everyone possesses by tweaking our perception. The success of this artist is due to the fact that his lithographs give the idea of a real figure’s components and their spatial oddness simultaneously [5].

When we look at a painting (generally at a 1 m distance), our attention moves from one point to another of the image. This means that our eyes are fixed on a point (fixation pause: time frame of 120 ms) and then move to another point until the entire image has been perused [6]. The identification of the scene gist is pre-attentive, i.e., within the fixation pause [4]; the interpretation of the different depth clues that contribute to the identification of a general model of the structures present in the painting relies on a huge set of memorized models of various familiar shapes, and can be considered unconscious [7]. The image of the entire painting appears simultaneously defined and detailed, though during a fixation pause, only the small zone surrounding the fixation point is perfectly clear. Outside this small zone, the image is blurred due to the undersampling of the retina periphery. At the end of the visual inspection, all images acquired during the fixation pauses are re-assembled on a mnemonic base, producing a well-defined categorization of the objects present in the painting.

In the case of paintings containing impossible structures, such as those created by Escher, we do not realize the three-dimensional inconsistency of the painting, because the re-assembling process does not take into account the spatial relationship among the acquired sub-images [8]. Still, we unconsciously perceive a sort of spatial incongruence challenging our attention [9,10]. Therefore, a more attentive parsing of the painting is necessary to detect the impossibility of the physical realization of the structures present in it.

In this paper, we demonstrate that, when we look at one of the Escher’s lithographs containing impossible structures, our visual system cannot work them out pre-attentively. We implemented a simplified version of a discrete model of the retina [11] whose output is a faithful representation of the image formed on the retina within a time fixation frame of 120 ms. This output is called pre-attentive retina image (PRI). The Belvedere lithograph was chosen as a stimulus subject. The results showed that, in a single glimpse (120 ms), an onlooker can realize the three-dimensional consistency of what she/he is looking at, only in a very spatially limited area of high resolution centered at the fixation point (visual angle ≈ 2.46°) with no clue of the three-dimensional consistency in the periphery of that area. Escher was aware of the structure and functioning of the human visual system; he challenged the onlooker with images characterized by components whose 3D inconsistency is unperceivable pre-attentively because they were drawn by the artist just outside the zone surrounding the fixation point. This skillful creativity is very probably the reason of his success. The main aspects of our paper from the point of view of image processing and image understanding are the following: (1) the peculiar and original digital filter to process the image, which simulates the human vision process, by producing a space-variant sampling of the image; (2) the software for the filter, which is homemade and created for our purposes. The filtered images resulting from the processing are used to understand impossible figures. As an example, we demonstrate how the impossible figures hidden in Escher’s paintings can be understood.

## 2. Anatomy of Retina

During the fixation pauses, scenes are formed on the retina, which is the innermost coat of the posterior part of the eyeball. The retina contains six different types of cells organized in three layers: cones and rods (photoreceptors), horizontal cells, amacrine cells, interplexiform cells, bipolar cells, and ganglion cells. Light transduction is performed at the level of the photoreceptors, and the output signals of the retina are carried to the brain by the optic nerve composed of retinal ganglion cell axons and glial cells [12].

The density of photoreceptors in the retina varies with topography (for details and figures, please refer to reference 12). In the fovea, a small depression in the center of the retina, there are basically only cones, densely packed (250,000 cones/mm^2^), with a center-to-center spacing of 2.5 µm. This distance set the limit of resolution imposed by the eye optics [13]. The fovea is used for high acuity tasks since each cone is connected with a single bipolar cell, which in turn is connected with a single ganglion cell. In the retina periphery cone, density decreases (11,500 cones/mm^2^), while rod density increases markedly to about 160,000 cones/mm^2^ and decreases to zero towards the far periphery [13]. In the far periphery, a ganglion cell collects the output of many cones, thus viewing a larger portion of the image and contributing more than the fovea to the scene gist recognition [4]. Therefore, the field of view can be split into a homogeneous high resolution area, i.e., the fovea, and a peripheral zone with decreasing resolution [12,13]. In analyzing images, the primate visual system functions on a compromise basis; i.e., it uses a data reduction scheme that preserves high resolution at the center of the fixation point, where the fovea is located (visual angle ≈ 1°), and reduces the resolution in the sensorial peripheral zone in function of the distance from the center of the fovea, i.e., in function of its eccentricity (visual angle > 1°). In this way, a wide field-of-view, high spatial resolution in the region of fixation and fast processing are simultaneously obtained [14].

## 3. The Model

We simulated the retina structure with a model in which the receptive fields (RFs) are positioned in a log–polar triangular lattice [11]. As the anatomical retina, the model is divided into two regions: the central zone (inside the green line, Figure 2a) and the peripheral zone (outside the green line, Figure 2a). In the central zone, which corresponds to the anatomical fovea, the RFs have all the same size, i.e., the size of a single cone; hence, spatial resolution is the highest achievable and uniform all over.

The peripheral zone corresponds to a zone with eccentricity higher than 1°. To build the RFs in this region, we traced a set of concentric rings with exponentially increasing radii with respect to ring index *h*, and divided the region into 2N equal sectors by means of 2N rays originating at the center of the fovea with a ray spacing angle of *π/N*. For each concentric ring, there are N RFs, which are built as circles, since the ganglion cell receptive field is more or less circular [15]. The center of each RF is calculated as the intersection between the odd or even rays with the odd or even rings.

The polar coordinates $\left(\rho_{h},\;\eta_{k}\right)$ of the even–even intersections are
(1)$$\rho_{h}=\rho_{0}{\left(1+\omega \frac{2\pi }{3N}\right)}^{2h}\;0\leq h$$
(2)$$\eta_{k}=\frac{\pi }{N}\left(\frac{2}{3}+2k\right)\;0\leq k<N$$
while the polar coordinates of the odd-odd intersections are
(3)$$\rho_{h}=\rho_{0}{\left(1+\omega \frac{2\pi }{3N}\right)}^{2h+1}\;0\leq h$$
(4)$$\eta_{k}=\frac{\pi }{N}\left(\frac{2}{3}+2k+1\right)\;0\leq k<N$$
where $\rho_{0}$ is the fovea radius (about 0.75 mm), and ω=0.9 is the RF overlap factor.

The radius (r) of each RF increases linearly as a function of its distance from the center of the fovea, i.e., as a function of its eccentricity [16] and is expressed in polar coordinates as
(5)$$r_{h}=\frac{2\pi }{3N}\rho_{h}$$

The spatial resolution in the peripheral zone decrease with the increase of both the eccentricity and the RF radius.

As a consequence of this construction strategy, the peripheral zone becomes a triangular lattice of circular RF, positioned alternatively in adjacent rings, i.e., each RF is a sampling point shifted half sample period between two consecutive rings. This model has a higher interpretative complexity compared to other models of retina based on rectangular or pseudo-rectangular lattices. Still it achieves the highest spatial coverage with the minimum overlapping of the RFs. This minimum overlapping is eliminated by replacing circular RFs with inscribed irregular hexagons, which are still biologically plausible [17], and allow a complete coverage of the peripheral zone (Figure 2a).

In the peripheral zone, there is an irreversible compression due to the multi-to-one correspondence of the RFs with the ganglion cells, since a single ganglion cell collects the output of many cones, thus viewing a larger portion of the image. The peripheral zone becomes a sort of variable spatial filter, less sensitive to fine details compared to the fovea.

In our model, the value of the output of each irregular hexagon (i.e., each RF) is calculated as the Gaussian weighted sum of the gray level of all the pixels $\left(P_{j}\right)$ of the acquired image present in that hexagon:
(6)$${\mathrm{RF}}_{i}=\sum_{j}w_{ij}P_{j}$$
where the weights $w_{ij}$ are the values of a normalized Gaussian centered on the i^th^ RF. The standard deviation of the Gaussian is chosen to have 99% of its area inside the RF.

The output of our model is the image formed on the retina during a fixation pause of 120 ms, which we called the pre-attentive retina image (PRI).

To show how the model works, we simulated the observation of the Belvedere lithography from a 1 m viewing distance, taken as the average viewing distance for an onlooker in a picture exhibition [18]. The number of RFs in each concentric ring, i.e., the compression in the peripheral zone of the model, was experimentally selected (see Experiment #1). For visual clarity, the model shown in Figure 2a has N = 40; i.e., each concentric ring of the peripheral zone (i.e., outside the green line) contains 40 RFs. A particular of the Belvedere lithograph processed by the model is shown in Figure 2b, and the corresponding PRI is shown in Figure 2c.

## 4. Experiment #1—Matching the PRIs

### Method, Procedure, and Results

Thirty volunteer participants, between the ages of 20 and 40, with normal or corrected-to-normal vision, and normal color discrimination, took part in the experiment. The number of participants was chosen according to [4]. The protocol was approved by the local Ethic Committee, and all patients signed the required consent form before the study.

Stimuli were presented on a 24” HP W2408H monitor with a resolution of 1920 × 1200 pixels and a refresh rate of 60 Hz. The monitor was standing in a vertical position to better reproduce the real size of the Belvedere lithograph (29.5 × 46.2 cm). A head-chin rest reduced participants’ head movements and ensured a constant viewing distance of 1 m. We chose a 1 m viewing distance since this is the average viewing distance for observing paintings of this size in an exhibition [18], but any other distance would work as well, since our demonstration is valid in all cases, no matter the distance.

Before the trial, a fixation cross on a gray background was shown for 1 s. Participants were instructed to press a next button to initiate the trial. The first stimulus presented for 120 ms was the Belvedere original lithograph, downloaded from www.epsilones.com. The fixation cross was then presented again for 1 s, followed by the second stimulus, i.e., the PRI of the lithograph obtained by our model, presented for 120 ms. The participants were asked to answer “Yes” or “No” to state whether or not they perceived the second image as matching the first one. This means that the participants had to compare the PRI produced by their visual system with the PRI produced by the model.

The procedure gradually reduced the spatial resolution of the PRI produced by the model by reducing the number of the RFs in each peripheral ring one at a time, until the participants recognized a difference with the first stimulus, reaching their threshold, i.e., the resolution limit set by the number (N) of RFs in each ring of the peripheral zone of the last PRI matching the Belvedere original lithograph.

If the participant doubted his/her choice, he/she could repeat the procedure until he/she felt at ease with the selected threshold.

All the threshold values (N) resulting from the experiment were averaged, obtaining a mean of 80.71 ± 3.1 as S.D. Eighty was the closest integer to the average, and was chosen as the number of RFs in each concentric ring of the model. As a consequence, the radius of the RFs in the first line of the peripheral zone was about 0.4 mm.

## 5. Experiment #2—The Angle of the Pre-Attentive Perception of 3D Incoherence

### Method, Procedure, and Results

The experimental conditions were the same as the first experiment. Before the trial, a fixation cross on a gray background was shown for 1 s. Participants were instructed to press a next button to initiate the trial. The stimulus presented for 120 ms was the devil’s pitchfork, an impossible structure also known as Schuster’s conundrum, 16 cm long [19]; the fixation point was positioned at the base of the tines (Figure 3a). The participants were asked to respond “Yes” or “No” to say whether or not they recognized the 3D incoherence of the image. The test gradually reduced the length of the tines until the participant could detect its 3D incoherence pre-attentively, reaching their threshold, i.e., the visual angle that contains the entire pitchfork image (Figure 3b).

All the visual angles resulting from the experiment were averaged, obtaining a mean value of 2.46° ± 0.09°, which corresponds to a window with a 43.0 mm radius with ±1.6 mm as S.D., at which 3D incoherence could be pre-attentively perceived. In our model the radius of this window approximated to the 42.80 mm radius of the 38th ring, corresponding to a visual angle of 2.45°.

The PRI of Figure 3a is shown in Figure 3b. It clearly demonstrates that, with a single glimpse, it is impossible to perceive 3D incoherence in the image since the tines outside the window have a very low spatial resolution. Therefore, it is necessary to move the fixation point along the tines to produce more than one PRI, whose assembling is mandatory to perceive the spatial incoherence of the whole. Figure 3d shows the PRI of Figure 3b. In this case, the pitchfork is entirely represented in the window of a 42.80 mm radius with a minimal loss of spatial resolution (the radius of the RFs of the 38th ring is 1.0 mm); hence, the 3D incoherence of the object can be pre-attentively detected.

## 6. Experiment #3—Looking for Incoherence in the Belvedere Lithograph within the Window of Pre-Attentive Perception

### Method, Procedure, and Results

The experimental conditions were the same as the first experiment. The stimulus was the image of the Belvedere lithograph overlaid with a window of a 42.80 mm radius, showing only the central portion of the image and blocking all the peripheral information with a gray layer (Figure 4a). After 120 ms, the window was obscured and the participants could change its position by means of the keyboard. A new window was then displayed for 120 ms, and the participants were asked to say whether they could recognize some spatial incoherence within the window. No incoherence was pre-attentively detected by the participants. As in the case of Experiment #2, the PRI (Figure 4b) would not have allowed the perception of any 3D incoherence due to the minimal loss of spatial resolution within the visual angle.

## 7. Experiment #4—The Window of 3D Incoherence in the Belvedere Lithograph

### Method, Procedure, and Results

The experimental conditions were the same as the previous experiments. The image of the Belvedere lithograph was overlaid with a window of a 43.00 mm radius, corresponding to a visual angle of 2.46°. The window was initially positioned on the zone of the interposition between the upper and lower level of the building where 3D incoherence is present. All the peripheral information was blocked. The participants could freely move the window over the lithograph and gradually increase the window radius by a 0.5 mm step by means of the keyboard until they perceived some spatial incoherence in a free viewing mode (i.e., with more than one glimpse).

The window that allowed for the perception of 3D incoherence had an average radius of 51.0 mm, with ±2.0 mm as S.D., corresponding to a visual angle of 2.92° ± 0.11° (Figure 5a). In our model, the radius of this window approximates to the 51.6 mm radius of the 46th ring, which corresponds to a visual angle of 2.95°. Figure 5b shows the corresponding PRI image of the Belvedere lithograph. This image clearly demonstrates that an onlooker looking at this lithograph cannot pre-attentively perceive the impossible cuboid inserted by Escher in the image since the spatial resolution outside the limit of pre-attentive perception (2.46°) is too low (the radius of the RFs belonging to the 46th ring is 2.85 mm).

## 8. Discussion

In this paper, we demonstrated that, when we look at one of Escher’s lithographs containing impossible structures, we cannot work out their 3D incoherence pre-attentively (i.e., within a time frame of 120 ms), because the artist has inserted these structures just outside a definite and limited zone surrounding the fixation point. This means that these artworks need many successive fixations to be wholly inspected, followed by a mnemonic process of re-assembling of all the sub-images acquired by the brain. Notwithstanding, 3D incoherence cannot be perceived yet, because image re-assembling does not take spatial relationship into account; however, a sort of spatial incongruence is unconsciously perceived [8,9,10]. Therefore, Escher’s lithographs provoke our intelligence by tweaking our perception of dimensions and push our mind to discover more beyond the first observation, challenging our concepts of the real world. This unsettling perceptual state forces the attention to a more painstaking examination of the lithographs to sort out contradictory details and realize how improbable the entire scene is. To our knowledge, the demonstration of how 3D incoherence is comprehensible only after a detailed parsing of his artworks is lacking, though there is a lot of literature on Escher’s works.

The output of the discrete model of the retina we implemented is the image formed on the retina within a time fixation frame of 120 ms (PRI). Experiment #1 allowed us to establish which PRI produced by the model has an RF lattice more similar to that of the PRI produced by the visual system. We tested the model on the image of the Belvedere lithograph viewed from a 1 m distance. Figure 2 shows an example of a PRI in which, for the sake of clarity, the number of RFs is much lower (N = 40) than that resulting from the experiment (N = 80).

Experiment #2 allowed us to establish the amplitude of the critical angle of pre-attentive 3D incoherence perception. We tested our retina model on the devil’s pitchfork image viewed from a 1 m distance (Figure 3a). Figure 3c shows the PRI of Figure 3a. The tines are long and merged due to the size of RFs in the periphery of the model (2.85 mm), so it is impossible to perceive the ambiguity of the image. Reducing the length of the tines until it is possible to perceive the ambiguity of the image in a single glimpse allows for a determination of the critical angle (≈2.46°). Figure 3d shows the PRI of Figure 3b: in this PRI, the tines are still well defined because of the size of RFs (1.0 mm) within the critical angle of view.

Experiment #3 allowed us to check the Belvedere lithograph for pre-attentively perceivable incoherence within the visual critical angle determined by Experiment #2. The PRIs of Figure 4a,b (positioned on the zone of the interposition between the upper and lower level of the building where 3D incoherence is present) confirmed that any incoherence presented in this zone would have been recognized because of the minimal loss of spatial resolution.

Experiment #4 allowed us to find the distance from the fixation point at which Escher inserted 3D incoherence structures in the Belvedere lithograph. The resulting window, obtained in free viewing mode, has a radius of about 51 mm, which corresponds to a visual angle of about 2.92° (Figure 5a), greater than the critical angle of pre-attentive 3D incoherence perception (2.46°). Figure 5b shows the corresponding PRI image, which confirms that the spatial resolution in the area between 2.46° (cyan line) and 2.92° is too low, i.e., the radii of the RFs are about 2.8 mm, and the under-sampled details in the lithograph appear merged and fuzzy.

The distance of observation of a lithography is not relevant in the analysis. The size of the lithography does not change with the distance, while the visual field increases as distance increases due to geometrical constraints; consequently, the area sampled by each RF increases, which will be located at an increasing distance with respect to the axis of the visual field.

This means that, at a 1 m distance, an RF that samples, for example, an area ***A*** belonging to the frame of the lithography, at a greater distance of observation, will sample a greater area, located at a greater distance from the visual field center, and belonging to the wall around the lithography. At this greater distance of observation, the area ***A*** will be sampled by an RF located closer to the visual field center, which will undergo an increase of its sampling area up the size of ***A***. Decreasing the distance (i.e., 0.5 m), the opposite occurs. As a consequence, no matter the distance of observation of the lithography, outside the pre-attentive perception window (cyan line), the model will produce identical PRIs. As shown in Figure 5c (observation distance: 0.5 m) and Figure 5d (observation distance: 2 m), the resolution is as low as it is in Figure 5b and does not allow any 3D incoherence perception.

These geometrical considerations are taken into account by Equation (5), which shows the constancy of the ratio between the RF radius, $r_{h}$, and the eccentricity, $\rho_{h}$ (distance from the center of the visual field).

Figure 6a–d show the PRIs produced by the model on the entire Belvedere lithograph viewed at four different distances, i.e., 0.5, 1, 2, and 3 m. The four PRIs are indistinguishable from each other, since the spatial resolution outside the pre-attentive perception window (cyan line) is comparably too low to allow any 3D incoherence perception.

In our opinion, Escher knew very well the structure and functioning of the human visual system. In fact, he challenged our mind with 3D inconsistent objects that overflow and defy our conceptions of the border of reality. In a single glimpse, it is impossible to perceive the spatial incoherence hidden in his artwork because Escher very cleverly drew it just outside the zone surrounding the fixation point, making this ability his own seal [20].

## Figures and Tables

**Figure 1 jimaging-06-00005-f001:**
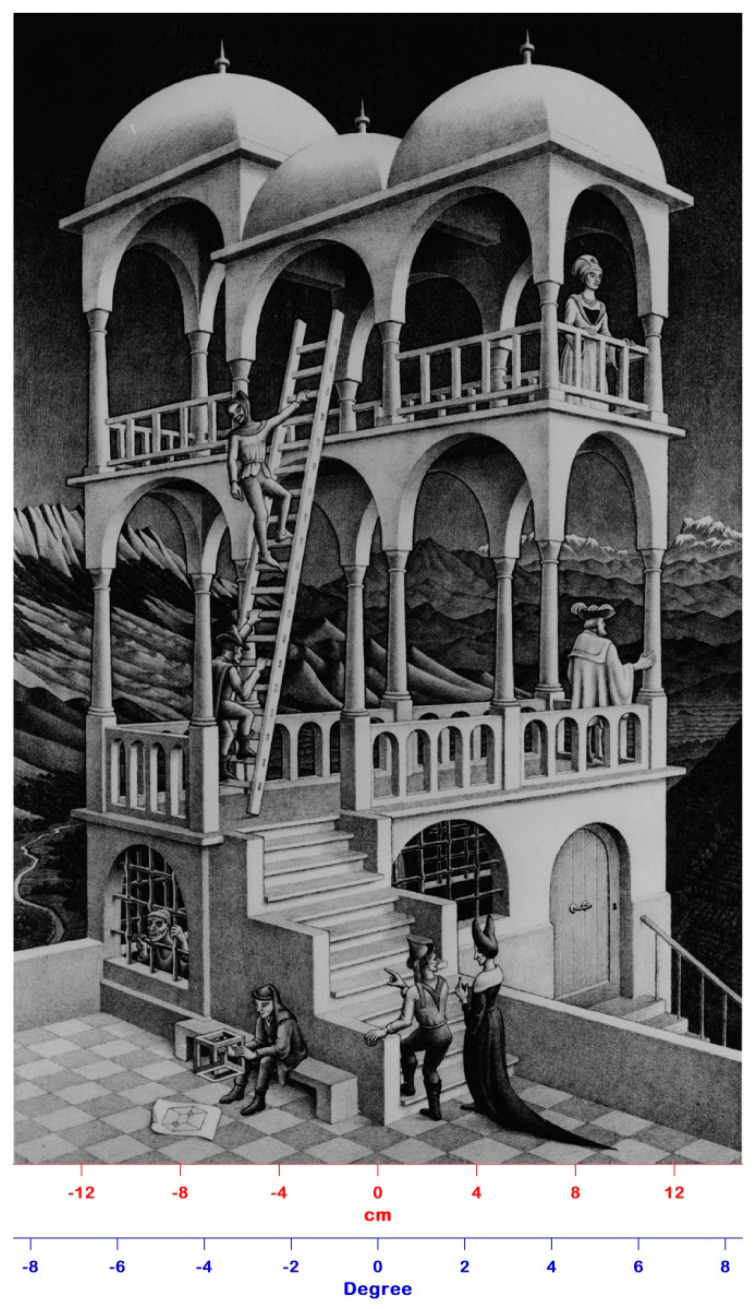
The Belvedere lithograph realized by Escher in 1958. The original dimensions of the artwork are 29.5 cm × 46.2 cm, which correspond to a visual angle of about 9° when the lithograph is viewed at a 1 m distance.

**Figure 2 jimaging-06-00005-f002:**
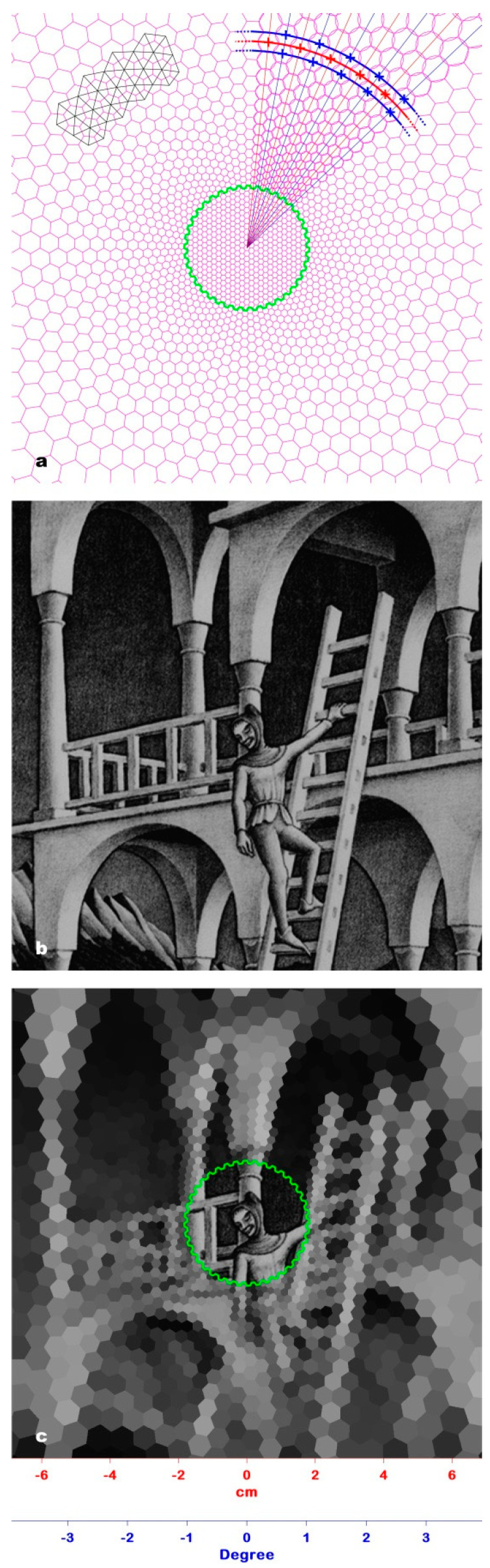
(**a**) Log-polar triangular lattice of the RFs in the retina model. The right upper part shows the building elements of the model: rays (blue lines) and rings (thick blue and red lines). The intersections between the rays and the rings mark the centers of the RFs (asterisks). RFs’ overlapping is minimized by replacing circular RFs with the corresponding inscribed irregular hexagons. The lattice triangles result from the joining of RF centers placed on successive rings (left upper part, black line). The serrated green line marks the border between the central zone and the peripheral zone. For the sake of clarity, N is set to 40. (**b**) Detail of the Belvedere lithograph chosen to test the retina model. (**c**) The pre-attentive retina image (PRI) of the Belvedere detail shown in Figure 2b. The serrated green line marks the border between the central zone and the peripheral zone.

**Figure 3 jimaging-06-00005-f003:**
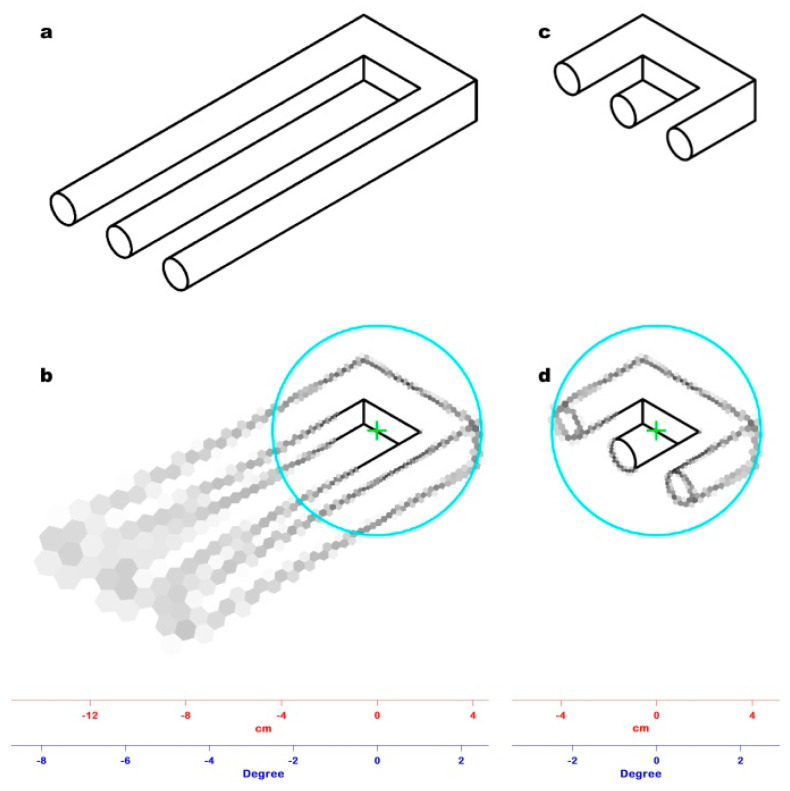
(**a**) The impossible figure used as stimulus in Experiment #2, known as the devil’s pitchfork or Schuster’s conundrum. (**b**) The result of Experiment #2: the tines of the fork have been reduced to the point at which 3D incoherence can be pre-attentively perceived, identifying the corresponding visual angle (2.45°). (**c**) The PRI of the devil’s pitchfork shown in Figure 3a: The fixation point (green cross) was positioned at the base of the tines. The cyan circle marks the limit of the window (42.80 mm radius) in which 3D incoherence can be pre-attentively perceived. Outside the window, the spatial resolution drops, hindering the detection of the image oddity. (**d**) The PRI of the trimmed devil’s pitchfork shown in Figure 3b: the fixation point (green cross) has the same position of Figure 3c. The entire impossible figure is represented within the window (cyan line) in which 3D incoherence can be pre-attentively perceived with minimal loss of spatial resolution, so its oddity is perceived in a single glimpse.

**Figure 4 jimaging-06-00005-f004:**
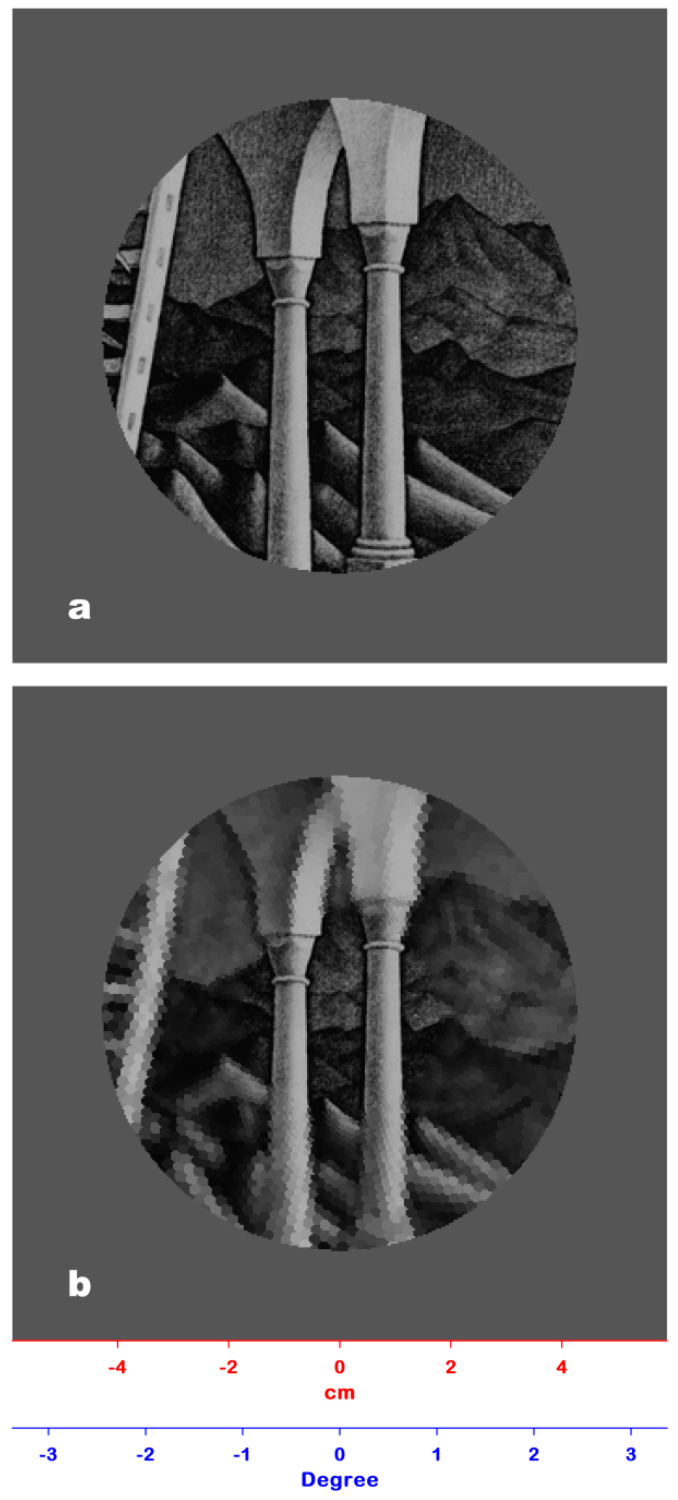
(**a**) The stimulus used in Experiment #3: The Belvedere lithograph overlaid with a window of a 42.80 mm radius showing only a detail of the artwork and blocking all the peripheral information. For the sake of clarity, only a part of the lithograph is shown. (**b**) The PRI of Figure 4a: Since the loss of spatial resolution within the window is minimal, any 3D incoherence present herein would have been pre-attentively perceived.

**Figure 5 jimaging-06-00005-f005:**
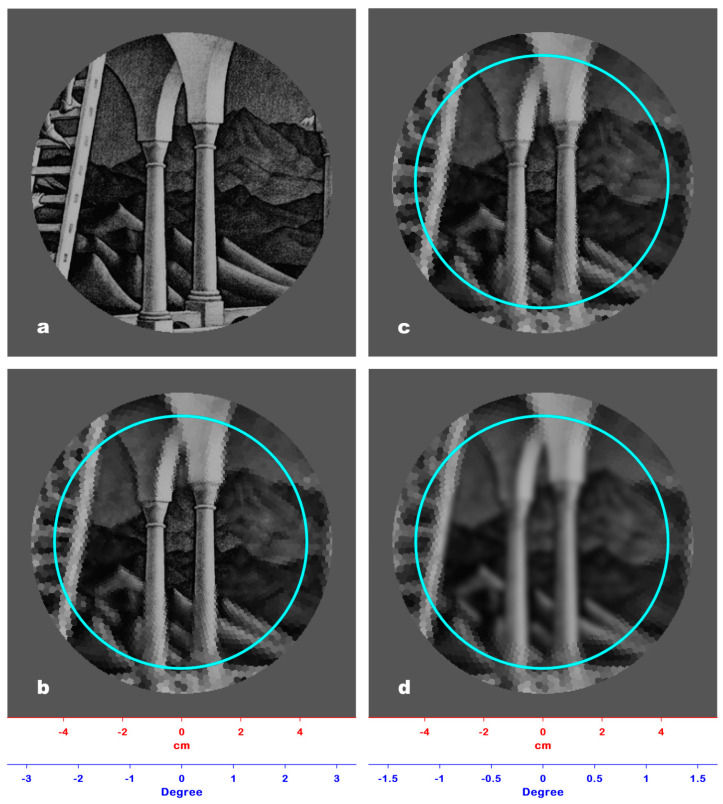
(**a**) The result of Experiment #4: The Belvedere lithograph is overlaid with a window of a 51.6 mm radius, which allows the perception of 3D incoherence in a free viewing mode at a 1 m observation distance. (**b**) The PRI of Figure 5a: The spatial resolution outside the limit of the pre-attentive perception window (42.80 mm, cyan line) is too low to allow the perception of 3D incoherence. (**c**) The PRI of Figure 5a viewed at a 0.5 m distance: At this distance, the perception of 3D incoherence is also hindered by the low spatial resolution outside the pre-attentive perception window. (**d**) The PRI of Figure 5a viewed at a 2 m distance: The spatial resolution outside the pre-attentive perception window is still too low to allow perception of any incoherence.

**Figure 6 jimaging-06-00005-f006:**
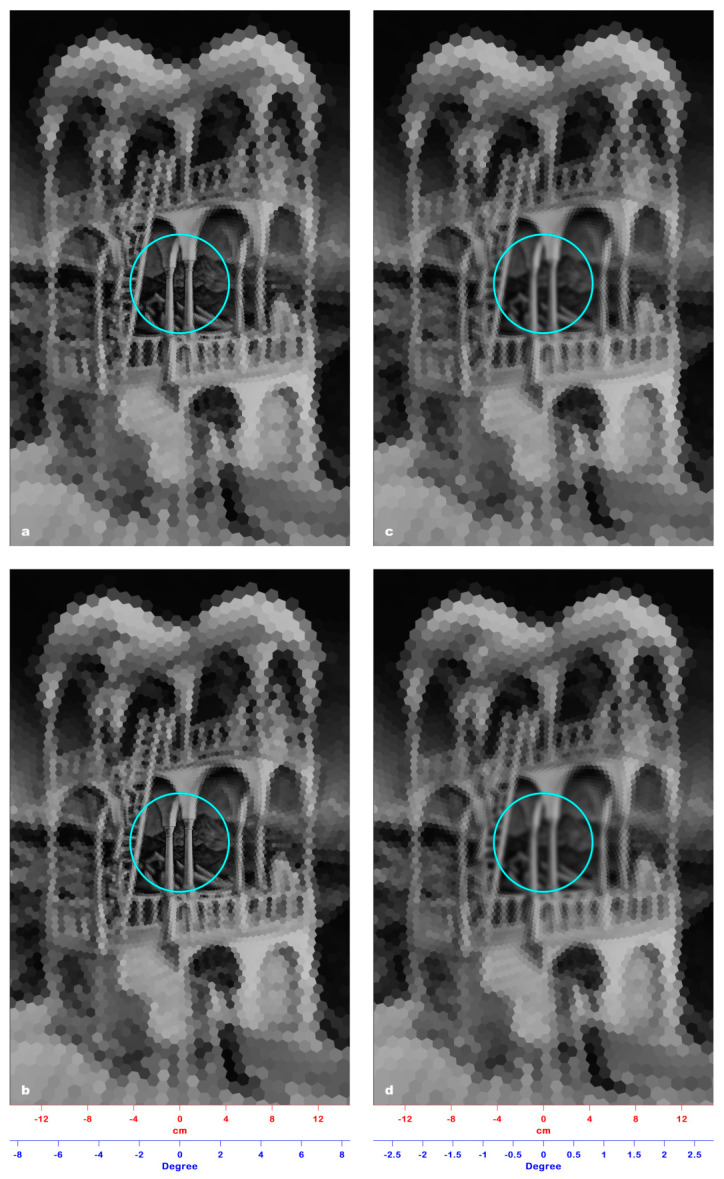
(**a**) The PRI of the entire Belvedere lithograph viewed at a 0.5 m distance. (**b**) The PRI of the entire Belvedere lithograph viewed at a 1 m distance. (**c**) The PRI of the entire Belvedere lithograph viewed at a 2 m distance. (**d**) The PRI of the entire Belvedere lithograph viewed at a 3 m distance. In the four images, the cyan line sets the limit of the pre-attentive perception window, outside of which the spatial resolution is comparably too low to allow any incoherence perception, making the four PRIs indistinguishable from each other.

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
