# Peer review of "Unveiling the Secrets of Escher’s Lithographs"

_2313-433X, 2020, doi:10.3390/jimaging6020005_

Round 1
Reviewer 1 Report
The authors have addressed an interesting and important issue of perceptual sciences including some aesthetical works by analyzing (and simulating) the pre-attentive processing of Escher’s lithographs showing impossible 3D figures. The whole methodological approach is high quality and the succession of 4 experiments is compelling. I only missed some essential papers on visual illusions and about aesthetic processing, but in the end incorporating these sources is quite minor. Some points to be addressed: 1) Illusions can be entertaining, they can even be artistic or called art such as in the case of Escher; they are mainly, however, and brilliantly shown by the present paper, a powerful tool to gain interest in perceptual sciences and to understand human perception. Carbon, C. C. (2014). Understanding human perception by human-made illusions. Frontiers in Human Neuroscience, 8(566), 1-6. doi:10.3389/fnhum.2014.00566. This should be made very clear at the beginning; they are real “tools” for systematic investigations. And they even make fun. 2) Escher is challenging as there is no clear solution; and if we find one in processing it quite locally, we will observe a moment later that this solution does not work out for a more global view on it: it is in the best sense of the word CHALLENGING ART which can be very interesting, and even being appreciated very much, especially for a very long time—see Belke, B., Leder, H., & Carbon, C. C. (2015). When challenging art gets liked: Evidences for a dual preference formation process for fluent and non-fluent portraits. PlosOne, 10(8), e0131796. doi:10.1371/journal.pone.0131796 3) Escher probably plays most effectively the card of ambiguity, or how we call it: “Semantic Instability” (SeIns): Although many researchers in the field of aesthetics claim that fluency is the main driver for becoming appreciated, higher forms of art, and Escher is a great example for it, need a kind of SeIns which is never fully solved: the interesting thing with Escher is that it is a infinite process of getting a clear message out of it, to “solve” the riddle behind; but perceptual factors as shown by the authors prohibit the full view on it. So SeIns could be an ideal theoretical background to explain the great appreciation people have for Escher. Muth, C., & Carbon, C. C. (2016). SeIns: Semantic instability in art. Art & Perception, 4(1-2), 145-184. doi:10.1163/22134913-00002049; Muth, C., Hesslinger, V. M., & Carbon, C. C. (2018). Variants of Semantic Instability (SeIns) in the arts. A classification study based on experiential reports. Psychology of Aesthetics, Creativity, and the Arts, 12(1). doi:10.1037/aca0000113 Further, very minor points: 4) A short illustration of the cone distribution on the retina (section 2) would help to understand the specific relations found for the human visual system even better, especially for the non-expert reader who will love this paper, I am quite convinced. 5) Please add information on how the people were incentivized. 6) Please use DOTS instead of COMMAS as decimal separator—it’s now inconsistently used! 7) Please specify how vision (and color vision) was assessed 8) Please explain how the number of participants have been calculated beforehand (if done so; else please refer to typical studies of that kind which used similar sample sizes). 9) Please always add a space between numbers and units, e.g. “43.0 mm” 10) msec is not a standard unit; use ms instead Signed CCCAuthor Response
Please see the attachment.

Reviewer 2 Report
Report
Coltelli, Barsanti and Gualtieri: Unveiling the Secrets of Escher’s Lithographs
Despite some exposure, Escher has not received much serious interest in the art world. He tends to be classified under the category of Fantasy Art, which is not really accurate and reveals the difficulty art critics/curators have in coming to terms with his work. Escher’s appeal has chiefly been popular, as a curiosity, as well as among scientists/mathematicians. Enter Coltelli, Barsanti and Gualtieri from Science/Information Technology and Biophysics. It is unsurprising that, in this case, such interest should come from Italy, as we are used to fine scientific work from that country, especially in neuroscience, though here the area is rather Perceptual Psychology and the location not Parma but Pisa.
The reason for focus on Escher in the context of Psychology relates to the fact that visual illusion is central to much psychological work in the last few decades, exemplified in the Anglophone sphere by Gregory. Gregory may be right in thinking that phenomena of illusion can provide critical insights into the workings of the visual brain. The Gibsonian riposte to this would be that, while mistakes occur in the real world, visually misleading cases are largely a laboratory phenomenon. However, this is not to deny that such cases are intriguing. The three authors want to do something they claim (to my knowledge correctly) not to have been done before, viz to put a numerical value on those glitches in Escher’s Impossible Pictures.
They begin with a computer-generated model of the visual experience of an Escher picture, viz the Belvedere. In fact probably no one really thinks the brain is digital in structure and operation. But the authors address this difficulty by their first experiment which compares the model with the real thing, postulating it as “biologically plausible”. They then go on to apply their method to a very simple case, the Devil’s Pitchfork, showing at which point in the fixation operation the glitch is visually apparent.
Experiments 3 and 4 apply this to the much more complicated example of Escher’s Belvedere. They choose an area to the right of the ladder on the first floor, explaining why the glitch is not apparent in a single fixation. The argument is that Escher cleverly puts the visual hiccup just out of reach of that fixation—and they give a numerical value to it. This seems convincing to me, though I would have chosen another point in the picture as a more immediate candidate. Of course a fixation can be located at any point. At the same time, as the example of Buswell’s Hokusai wave illustrated in the 1930s, a picture directs the eye (without compelling it). In which case I would suggest that Escher’s Belvedere leads the eye up via the stairs and ladder—which would make the right of the ladder a less obvious candidate for fixation than the left. Still, a visual experiment any reader may carry out shows that you cannot register both top and base of a given pillar in the same fixation. So the authors’ equations pass the test of everyday visual experience. Incidentally, the authors also demonstrate that perception of visual incongruity is not affected by viewing distance.
Having said all the above, I must declare that, maths not being my discipline, I leave verdict in this specific respect to another reviewer. For my part, I felt the article was lively and engaging and made its point. It should be published.
I note as an addendum that, though the article’s English is highly competent, there are minor errors of idiom which could be corrected, as follows:
—l.14: “behind” as used here is unidiomatic in English. Change to “unveil…hidden secrets” or “unveil what lies behind…secrets”
—l. 127: “respect to”. Change to “compared to” (“respect to”=idiom in Italian but not in English)
—ll. 185, 219, 236: in Italian we say “same of” (“stesso di”) but English says “same as”
—l. 226: change word order to “would not have allowed”
Finally, I state no objection to the three authors reading this report with my name attached.
